# Exploring General Intelligence of Program Analysis for Multiple Tasks

## Abstract

As the program scale ever grows, program analysis tasks become increasingly diverse and complex. Effective learning-based program analysis tasks need to consider many factors including not only program structures and semantics but also running platforms and compilation configurations. Existing works are oftentimes on a basis of source code or abstract syntax tree, but unfortunately lack sufficient program structure and semantic information to unveil program insights. To this end, we propose a new program analysis model that takes advantage of great power of graph neural networks from the level of assembly code, which can solve both traditional program semantics-based problems and compilation-related problems, by fusing the semantics of the program's instructions and existence of multiple graph structures in programs. We tested it on two program-level tasks and achieved as high accuracy as 82.58% and 83.25%, respectively.

## 1 Introduction

With the development of information technology, computer programs are used in an increasingly wide range of fields. As the number and variety of programs continue to expand, the task of analyzing programs becomes more significant and complex. Common program analysis tasks include program classification, duplicate code detection, making programming suggestions (syntax and algorithms), program vulnerability analysis, bug detection, cross-language program translation, code annotation, etc. Program analysis is not a simple problem and many factors bring complexity to it. First, program functions are not implemented in a unique way; apparently completely different codes may implement the same functions, and similar codes may implement quite different functions. Second, programs may be written in different languages and run on various platforms. The same code compiled by different compilation configurations may also have outstanding differences in terms of correctness and performance. All of these are the major factors to consider during program analysis.

In recent years, more and more researchers have introduced machine learning approaches to solve complex program analysis problems. Earlier approaches treat codes as shallow textual structures, e.g., sequences of tokens (Allamanis et al., 2016a; Hindle et al., 2016; Allamanis et al., 2016b; Pradel & Sen, 2018), and analyze them with natural language processing models. These approaches ignore the rich and explicit structural information that exists in a program. In addition, since many variables are defined and used not adjacent to each other or even far apart, this "def-use" relations are difficult to capture by this kind of model. However, such long-distance dependencies, can be reflected in the structural information of the program.

By abstracting programs as graphs and using graph neural networks (GNN) for program analysis tasks, better use can be made of these structure information that are readily available in programs. Graphs such as control flow graph, call graph, and data flow graph reflect the control flow, function calls and data flow relations of a program. Previous approaches based on graph neural networks have used only one graph structure (Xu et al., 2017; Ben-Nun et al., 2018; Wang et al., 2020). However, different graph structures respond to different program features, so it is necessary to combine multiple graphs together to obtain richer structural information.

Existing methods explore program embedding through different inputs, e.g., source code, abstract syntax tree, assembly code and so forth. Constructing embeddings via source code or AST makes good use of semantic information in program variables. However, they lack compile-time information and cannot solve compilation- and architecture-related analysis tasks. In order to build embed-

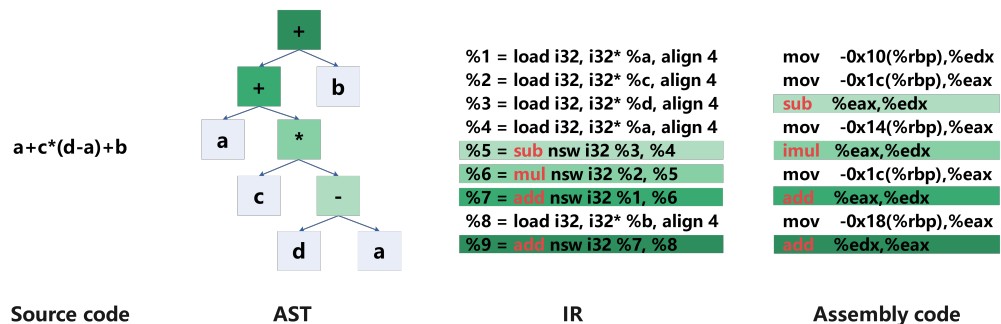

Figure 1: Different formats of program representation.

dings that can solve both source-code level tasks (program classification) and compilation-related tasks (program vulnerability analysis), we choose to build embeddings through the assembly code of a program. The embeddings we construct not only contain semantic and structural information of the program code, but also reflect differences in compilation configurations, so they can be applied to a wider range of tasks.

In summary, this paper makes the following contributions:

- We abstract a program into multiple graphs, extract different program structure information, and perform multi-dimensional analysis of the program.
- We build program embedding based on the assembly code, offering the possibility to analyze both source code related and compile related analysis tasks.
- We perform procedure-level and program-level analysis, capturing both local and global features.
- We measure on two program analysis tasks, program classification and binary similarity detection, and achieved an accuracy of up to 82.58% and 83.25%, 8.6% and 84.4% of improvement over the state-of-art approaches.

## 2 CHALLENGES OF PROGRAM EMBEDDING

### 2.1 MULTIPLE PROGRAM REPRESENTATIONS

From the perspective of language syntax, a program may be represented in many different formats, such as source code, abstract syntax tree (AST), compiler IR, assembly code, etc. Fig. 1 shows the four formats with one code example. As seen from Fig. 1, different formats share similar information but also have their unique characteristics and complexities. For example, only the source code and AST contain variable literals. Therefore, which level of code to embed with deserves a serious study.

Program embedding from source code certainly has its own merits. A programmer uses English words with clear meaning for variable or function names when coding. Direct embedding over these names can capture the literal semantics for program analysis. Also, source code is small sized, often within a few kilobytes. However, the downside is having to maintain a vocabulary for all possible variable names, which can be an incredibly large number considering programmers can name variables as they want. Moreover, embedding over the source code lacks information with respect to program structure semantics.

Another line of program embedding is over AST, an abstract representation of program syntactic structure in the form of tree. Unfortunately, embedding over AST also suffers from the problem of oversized vocabulary lists and the shortage of program structure information.

This work chooses to embed a program based on the assembly code for several reasons. An assembly code can be compiled from a source code. It is specific to a processor architecture, for example x86-64. It is typically less stylish and tight to program semantics. For example, programs that are syntactically different but semantically equivalent tend to correspond to similar assembly codes.

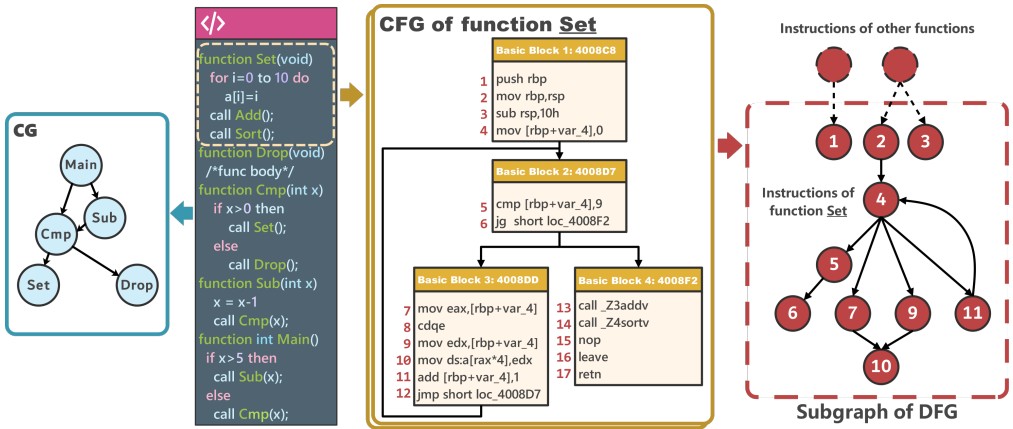

Figure 2: A CFG, CG and DFG of a program.

Moreover, the assembly code often embraces more compilation-specific characteristics than the program intermediate representation or AST (I. Neamtiu & Hicks, 2005). This work aims to solve the task of binary similarity detection, so it is necessary to distinguish between different architectures and compilers. This compilation-related information is not available in the source code or AST. Therefore, embedding over the assembly code can solve both source code related problems such as program classification, and compilation-related problems such as binary similarity detection.

## 2.2 CHALLENGES BY NUMEROUS GRAPHS OF A PROGRAM

There are plenty of graph structures implicitly inside a program, including but not limited to control flow graph (CFG), call graph (CG), and data flow graph (DFG) and more. A CFG is the graphical representation of the control flow of a code, mostly used in static analysis. It shows all execution paths that can be traversed during a program execution. Edges in a CFG portray conditional branches and nodes represent basic blocks. It contains loops, branches, and other structures, which is difficult to obtain directly from the source code of a program. A CG represents calling relationships between functions. Each node represents a function and each edge $(a, b)$ indicates that the function $a$ is calling $b$. Thus, a cycle in the graph indicates recursive procedure calls. A DFG represents def-use chains in a program. A "def" means an assignment to a variable and a "use" is a reference. A program consists of a large number of ordered operations forming a partial-order graph. The data flow graph reflects data dependencies between these operations.

Different graphs represent quite different program structures and semantics. Program embedding requires to embed as much semantic information as possible. How to embed these graphs together in an effective and efficient way is a challenging problem. Furthermore, these graphs are relative large for the efficient graph-based deep learning models (e.g., graph neural networks) to run. As tested, the number of nodes and edges of the DFG can be as high as 10847 nodes and 26910 edges. These large-scale graphs incur tediously long training time using graph neural networks.

## 3 METHODOLOGY

In this section, we propose a program embedding methodology that embeds multiple graph structures in a program as a whole from the level of assembly code. For the best practice, we adopt the deep learning models of graph neural networks (Scarselli et al., 2009) to represent the semantics of these three types of graphs. The proposed methodology has two standalone training processes consisting of two models, *BERT+* and *Gated GNN*. We fine-tune the pretrained BERT model to derive the initial embedding of each node of a CFG or DFG. Since the BERT model is specifically fine-tuned for program understanding, we name the model as BERT+, to distinguish. The second model we use is Gated GNN (GGNN) to derive the whole embedding of the CFG, CG and DFG, as GGNN is good at memorizing long-term data dependencies between nodes.

Depending on the specific tasks, program analysis may be in either procedure or program level. Procedure-level analysis is usually conducted to analyze a program in the unit of procedure, therefore for local function-level tasks. Program-level analysis is for program-level tasks. In this work, we focus on two program-level analysis tasks of program classification and binary similarity detection. However, procedure-level analysis tasks such as function name prediction can also be solved by the proposed model, which we will leave as future work.

For procedure-level analysis, we construct a control flow graph for each procedure and combine it with instruction semantics to complete the embedding of each procedure. Then the embedding of each procedure is used to do procedure-level analysis. For program-level analysis, we represent the calling relations between different procedures by call graphs and represent the data dependencies between all variables in the program by data flow graphs. We embed every call graph and data flow graph for each procedure and combine the embeddings of all procedures to give rise to the embedding of the entire program.

## 3.1 GRAPH NEURAL NETWORKS

The objective of Graph Neural Network is to learn the node representation and graph representation for predicting node attributes or attributes of the entire graph. A *Graph Neural Network (GNN)* (Scarselli et al., 2009) structure $G = (V, E)$ consists of a set of node $V$ and a set of edge $E$. Each node $v \in V$ is annotated with an initial node embedding by $x \in \mathbb{R}^D$ and a hidden state vector $h_v^t \in \mathbb{R}^D$ ($h_v^0$ often equals to $x$). A node updates its hidden state by aggregating its neighbor hidden states and its own state at the previous time step. In total, $T$ steps of state propagation are applied onto a GNN. In the $t$-th step, node $v$ gathers its neighbors' states to an aggregation as $m_v^t$, as shown in Eq.1. Then the aggregated state is combined with node $v$'s previous state $h_v^{t-1}$ through a neural network called $g$, as shown in Eq.2. $f$ can be an arbitrary function, for example a linear layer, representing a model with parameters $\theta$.

$$m_v^t = \sum_{(u,v) \in E} f(h_v^t; \theta) \ (1) \qquad h_v^t = g(m_v^t; h_v^{t-1}) \ (2) \qquad h_v^t = \mathbf{GRU}(m_v^t; h_v^{t-1}) \ (3)$$

*Gated Graph Neural Network (GGNN)* (Li et al., 2016) is an extension of GNN by replacing $g$ in Eq.2 with the *Gated Recurrent Unit (GRU)* (Chung et al., 2014) function as shown in Eq.3. The GRU function lets a node memorize history long-term dependency information, as it is good at dealing with long sequences by propagating the internal hidden state additively instead of multiplicatively.

## 3.2 PROCEDURE-LEVEL ANALYSIS

A procedure is a basic unit that implements a specific function. We construct embeddings for all procedures separately. The control flow of procedures and the semantics of instructions are the key features. We construct control flow graphs for all procedures and use the semantic information of instructions as the initial embedding of nodes in the graph.

The following describes how to build procedure-level embedding from assembly code.

Assembly code is a flat profile of instructions, $Ins_i(i \in 1 \ldots m)$. Each instruction $Ins_i$ is composed of a series of tokens, $t_i(i \in 1 \ldots n)$. These tokens are of many types, including operators and operands. Some instructions are compute operations with respect to register values (e.g., ADD, SUB). Some move values between registers and memory (e.g., LOAD, STORE). Others represent conditional branch or jump to other locations (e.g., JZ, JMP).

A basic block $B_i(i \in 1 \ldots K)$ is composed of a sequence of instructions without any control flow statements. A basic block has only one entry and one exit and may have multiple following basic blocks. When the program pointer jumps from a source to a target block, we connect an edge from the source to the target. Viewing the basic blocks as nodes and the connections as edges, we form a CFG. For instance, for x86 direct branches, there are only two possible target blocks for a given source block, which we can refer to as the *true block* and *false block*. Fig. 2 shows an example of CFG (the yellow part). The program in Fig. 2 has several procedures, and we show the CFG of Procedure Set. The loop structure of this procedure is clearly shown in the figure.

The semantics of instructions is crucial for program analysis. We use the semantics of each basic block to construct the initial embedding of each CFG node. We consider the instruction token

sequence of a basic block as a sentence and use the BERT (Devlin et al., 2018) model to embed them. In natural language processing problems, BERT is a widely used method for producing sentence embeddings. Here, we borrow BERT to obtain the semantic relations between different tokens.

The embedding of a basic block should recognize several goals, including the affinity relation between words in a sentence, affinity relation between basic blocks, and implicit meaning of compilation configurations. To achieve the three goals, we train the pre-trained BERT model by three tasks separately. The input of the BERT model is the token sequence of a basic block and output is an embedding vector. In the first task, we learn the capability of exploring the affinity relation between tokens in a basic block so as to check the missing tokens. In doing so, we mask out some tokens randomly given a token sequence and predict the masked out tokens by comparing the output embedding by the BERT model and the ground-truth embedding. Secondly, the model learns to recognize the affinity relation between basic blocks, i.e., predicting whether two basic blocks are adjacent in the sense that the corresponding nodes are connected by edges in a CFG. For training, we randomly generate a large number of basic-block pairs. If two basic blocks are adjacent to each other, the corresponding pair is labeled as true and otherwise false. This task allows the embedding to know the execution context of a basic block. Thirdly, the model learns the capability to check the compilation configuration by determining whether two basic blocks come from the same compilation configuration. As a result, the embedding contains the semantic information of compilation configurations. This capability is especially useful in the task of binary similarity detection, where distinguishing between different compilation configurations is needed. For training, we randomly generate a large number of basic block pairs as samples. If a pair is compiled from the same configuration, a sample is labeled as true and false otherwise.

As shown in Fig. 3, once we have embedded every basic block, we can build the embedding of the whole graph, which is the embedding of this procedure, through graph neural network. Once the embeddings of all procedures are obtained, we complete the procedure-level analysis.

### 3.3 PROGRAM-LEVEL ANALYSIS

While procedure-level analysis provides us program local semantics, how these procedures are compounded to implement highly complicated functions needs to be understood with the help of program-level analysis. In doing so, a call graph and a data flow graph that characterize the global view of program analysis are studied.

A call graph reflects calling relations between procedures. Each node denotes a procedure and is connected by directed edges, pointing from caller to callee. The call graph clearly provides the execution context from one procedure to another and propagates the control and data semantics across procedure boundaries. Hence, it is more than needed to analyze inter-procedural program analysis problems such as memory inefficiency detection (Chabbi & Mellor-Crummey, 2012) and program slicing (Gallagher et al., 1991). As shown in Fig. 2, the procedures `Main`, `Sub`, `Cmp`, `Set`, and `Drop` are represented by nodes, and the edges between them indicate the calling relations.

To embed the entire call graph to form a program-level embedding, we initialize the node embedding of a CFG with the procedure embedding previously generated by CFG. Then we propagate the semantics of the embeddings of every node with the GGNN graph propagation algorithm, and finally generate an embedding, denoted as $g_{cg}$.

While a call graph provides control flow information, a data flow graph presents program execution flows of associated data value snapshots. We construct a data flow graph from the assembly code. An assembly code consists of a great number of instructions. A single instruction executes a single step of data processing, such as computation, data load, data store, etc. Each instruction has one or more operands. An operand is defined by one instruction such as loading from memory by the `LD` instruction, computed from other variables by `MOV`, and used by other instructions such as `ADD` or storing to memory by `ST`.

We treat an instruction as a node. When a variable is defined by Instruction $I_a$ and used by Instruction $I_b$, we connect a directed edge from $I_a$ to $I_b$. Consequently, we obtain a data flow graph of the program. It reflects the data dependencies between instructions of the program and the "def-use" chains. Fig. 2 shows a graph of DFG. Each node represents an instruction, the nodes in the dashed boxes are instructions belonging to the `Set` procedure, and the ones outside the boxes represent

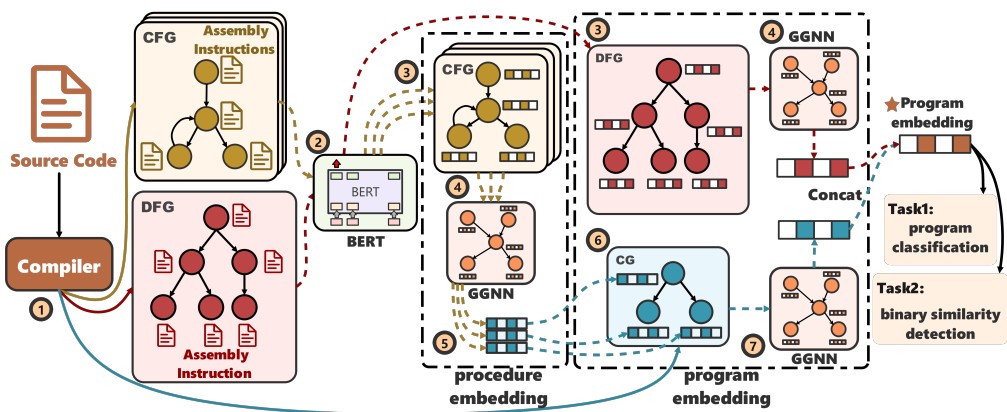

Figure 3: The process of program embedding.

instructions of other procedures. Relative to CFGs and CGs, DFGs are much larger. By analyzing a small program dataset, POJ-104 (Lili Mou, 2016), more than 53% of DFGs have over 512 nodes and even more than 15% can exceed 1024 nodes. Following the training methodology for generating initial embedding for basic blocks in a CFG, we construct the initial embedding of each node in a DFG using BERT by viewing the token sequence of an instruction as a sentence in the BERT model. Whether two instructions are adjacent or not is indicated by whether there is an edge between two nodes in a DFG. The data flow graph is also constructed as an embedding through GGNN, and this embedding is called as $g_{dfg}$.

As shown in Fig. 3, we construct a control flow graph for each procedure and obtain its embedding. Using the embedding of each procedure and the calling relations between procedures, we obtain the first program embedding $g_{cg}$. Afterwards, we construct the second program embedding $g_{dfg}$ using the data flow relations in the program. Putting together these two embeddings, a single final program embedding is derived for the target program (Eq.4). Finally, we use this program embedding to perform various types of program analysis tasks.

$$g_{prog} = \textbf{Dense}(g_{cg} \parallel g_{dfg}) \quad (4)$$

## 4 RESULTS

### 4.1 EXPERIMENTAL SETUP

**Benchmark.** We uses the POJ-104 (Lili Mou, 2016) dataset as our test data. It is collected from a Pedagogical Open Judge system. The dataset contains 104 program classes written by 500 different people. After pre-processing, we ended up with a total of 38083 programs. We compile these programs with two compilers and four compilation configurations (*GCC-O2*, *GCC-O3*, *LLVM-O2* and *LLVM-O3*) and use 70% of the obtained programs as training data, 15% as validation data and 15% as test data.

**Task.** To test the efficacy, we evaluate the proposed model on two tasks, program classification and binary similarity detection. Program classification is a commonly seen program analysis task. What specific functions a program implements is one of the most important features of a program. Whether the program embedding trained by the model can accurately perform the program classification task shows whether the model can capture the key information of the program.

For each program, we produce a label that indicates the category of program it belongs to. The labels range from 1 to 104, indicating each of the 104 categories of programs. Once the model has generated embeddings for each program, we use softmax to complete this multi-categorization task. If the predicted category is consistent with the label, the sample is correctly predicted. We use the prediction accuracy of all samples in the test set as the measure for this task.

Another task we tested is binary similarity detection. Binary code similarity detection tasks are often applied in problems such as plagiarism detection, malware detection, and vulnerability search (Xu

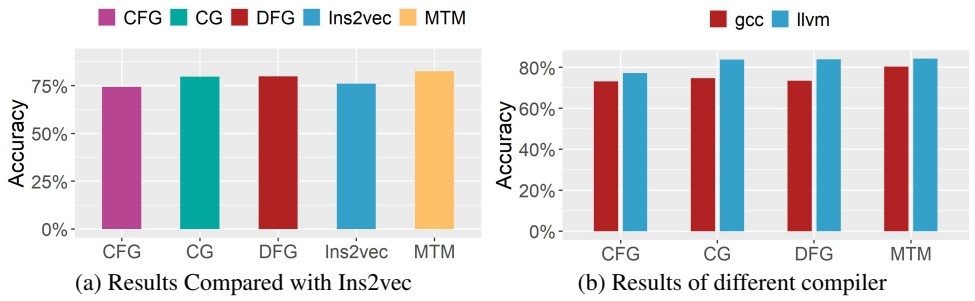

(a) Results Compared with Ins2vec    (b) Results of different compiler

Figure 4: Classification accuracy

et al., 2017). As in the vulnerability search problem, a vulnerability in one source code may spread across hundreds or more devices that have diverse hardware architectures and software platforms. In order to find issues from the same source code in different binary, we need to perform a binary similarity detection. Find the binary that belongs to the same vulnerability code.

In this task, we use two subtasks to help evaluate the model. The first subtask is cross configuration binary code detection. The same source code is compiled with different compilers and different compilation options. Our goal is to make sure the pairs from the same source code has higher similarity scores than pairs from different code. Same as (Yu et al., 2020a), we use the siamese framework to complete the training for the similarity determination task.

For this subtask, we use the similarity distance as a measure. Each training and testing sample consists of two binary. The model will predict whether they originate from the same source code or not. Pairs from the same source code are called positive pair, and pairs from different source codes are called negative pair. The average similarity distance of all positive pairs are called positive distance, and the average similarity distance of all negative pairs are called negative distance. The larger the difference between these two distances, the better the model's ability to differentiate. To obtain training and test data, we randomly generated 100k pair of samples. In half of the samples, the two programs are compiled from the same source code with different compilation configurations. In the other half, the two programs are from different source codes.

The second subtask is compilation classification. In this task we will classify the compilation options which binary belongs to. There are 4 categories in total, *GCC-O2*, *GCC-O3*, *LLVM-O2* and *LLVM-O3*. We also use softmax for this multi-classification task. The accuracy of the classification is used as a measure.

**Implementation.** The inputs to our model include three graphs CFG, CG, and DFG. A binary analysis tool called `angr` (Shoshitaishvili et al., 2016) is utilized by taking as input the binary code and constructing these graphs for every single program. To complete these two tasks, each input sample is labeled to indicate its category (category id, from 1 to 104) and how it is compiled (compilation option id, from 0 to 3). To complete the first subtask of binary similarity detection, each binary pair will have a label (1 for homologous / 0 for heterogeneous) indicating whether they come from the same source code. We call our model Multiple Tasks Model (MTM).

**Hyper-parameters.** We show only the values of all hyper-parameters used in the final experiments. The learning rate used is 0.0001 and batch size is 32. The number of steps of CFG, CG and DFG are 8, 4 and 8. The output dimension of CFG, CG and DFG are 128 2048 and 128. The hidden dimension of all three graphs is 128.

**Hardware.** All deep learning tasks were performed on 8 Nvidia Tesla V100 GPU (Nvidia-Inc., 2017) cards of the Volta architecture. Each has 5120 streaming cores, 640 tensor cores and 32GB memory capacity. The CPU host is Intel Xeon CPU 8163 2.50GHz, running Linux kernel 5.0. All the other non-deep learning tasks were run on the host.

### 4.2    PROGRAM CLASSIFICATION TASK

For the program classification task, we compared our model with Ins2vec (Ben-Nun et al., 2018). Ins2vec constructs a graph called XFG which incorporates data flow, and control flow information

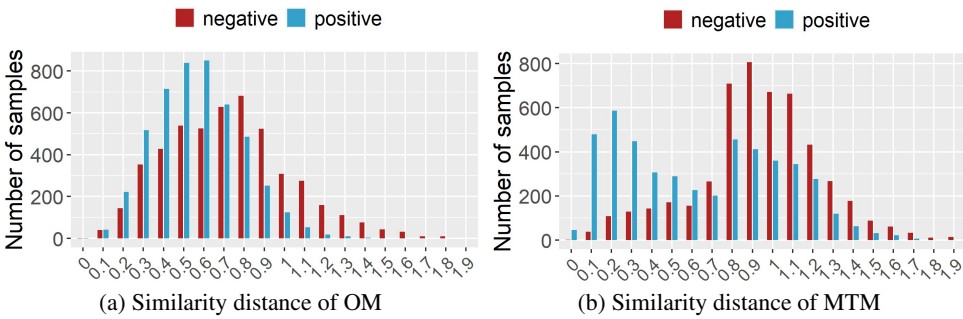

(a) Similarity distance of OM

(b) Similarity distance of MTM

Figure 5: Similarity distance of OM and MTM

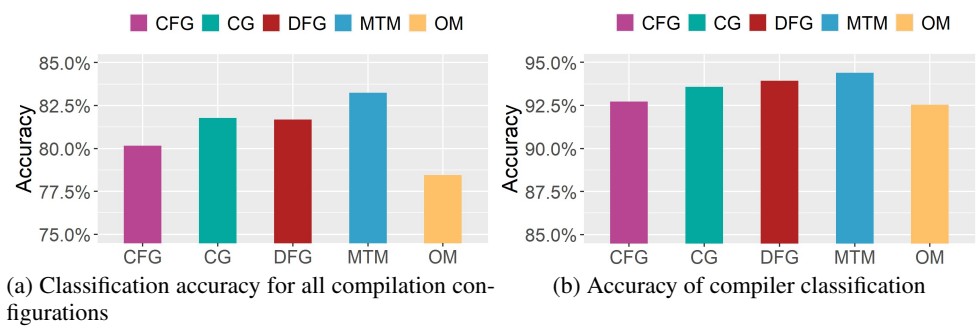

(a) Classification accuracy for all compilation configurations

(b) Accuracy of compiler classification

Figure 6: Compilation option classification accuracy

of the program. It takes the edges of the graph as the context of the nodes and uses the skip-gram model to get the embedding of each node. After that the embedding of the program is obtained with LSTM. The results of the comparison are shown in Fig. 4(a). Compared with Ins2vec, we improve the accuracy by 8.6%. This shows that our model has good accuracy in solving source code related tasks.

In addition, we show the effect of each submodel in this task separately. The DFG and CG submodels represent the use of $g_{dfg}$ and $g_{cg}$ as program embedding, respectively. The CFG model indicates that only the CFG embedding of each procedure is used. The embedding of the program is the average of the embedding of all its procedures. As can be seen from Fig. 4, when only the control flow information of the program is utilized, only 74.38% accuracy can be obtained. When combining the control flow information and procedure call information, the accuracy is improved to 79.77%. When only the data flow information of the program is extracted to construct the embedding, an accuracy of 79.93% can be achieved. From the above results, it can be summarized that added, multiple program structure information is an effective way to obtain a more accurate program embedding.

Fig. 4 (b) shows the prediction results of the model for different compilers. The model achieves 80.41% accuracy under GCC and 84.21% accuracy under LLVM.

### 4.3 BINARY SIMILARITY DETECTION TASK

For this task, we chose the state-of-the-art model *Order Matters Model*(OM) for comparison. OM generates a control flow graph for each program and uses GNNs to construct embeddings.

The binary similarity detection task is divided into two subtasks. In the first subtask, we need to determine whether two binaries come from the same source code. It is not easy to accomplish this task on POJ-104 dataset. Because the source code to complete the same POJ question may have a high degree of similarity, it will be more difficult to determine whether they originate from the same source code after being compiled with different compilation options. Fig. 5 shows the results of our tests. It can be clearly seen that the distribution of similarity distances of the positive pair is more

different from that of the negative pair under our model. The average positive and negative distance of OM is 0.5668 and 0.7272. The average positive and negative distance of MTM is 0.6394 and 0.9353. The difference between the positive distance and negative distance for OM is 0.1604, and the difference for our model is 0.2959. Our model has better differentiation capabilities.

The second subtask is the compilation classification task, which needs to classify the compiled configuration of the program. With this subtask, the ability of the model to distinguish between different compilation configurations can be judged. Fig. 6 shows the experimental results. When classifying among all 4 compiled categories, we obtained an accuracy of 83.25%. We also tested the ability of the model to discriminate between different compilers. All samples are divided into two categories, compiled by gcc and compiled by llvm, respectively. In this test we obtained an accuracy of 94.4%, as shown in Fig. 6 (b). The above results demonstrate that our model has good differentiation ability for different compilation configurations and can successfully perform compilation-related program analysis tasks.

## 5    RELATED WORK

In recent years, applying neural networks onto program analysis has attracted ever-increasing attention (Raychev et al., 2014; Hindle et al., 2016; Allamanis et al., 2016b; Pradel & Sen, 2018; Kanade et al., 2020; Feng et al., 2020). Token embedding for program representation using existing natural language processing (NLP) techniques serve as a good start. Hindle et al. (2016) proved that code was very regular and could be modeled by statistical language models. Allamanis et al. (2016b) treated the code as textual tokens and introduced an attention-based neural network to summarize functions. Pradel & Sen (2018) made use of the information conveyed by natural language elements in the source code, e.g., names of variables and functions, by representing them with Word2Vec for bug detection. Kanade et al. (2020) leveraged BERT to perform high quality source code embedding for program repair and showed that BERT is very effective in extracting the semantic information of code. Even though the above works have shown the efficacy of token embedding to some extent, they are short of structure-based semantic representation. Hu et al. (2018); Chen et al. (2018) initially considered ASTs based on a finding that program representation is different from traditional machine translation because the code owns structures. Alon et al. (2019a;b; 2020) represented a code snippet as the set of compositional paths in the AST and solved the problems of code completion, method name prediction and code documentation. There is a large body of works employing GNNs to mine program structure oriented semantics. Xu et al. (2017) explored the embedding of binary functions out of CFGs to detect cross-platform binary similarity by measuring the distance between the embeddings of two functions. Lu et al. (2019) attempted to fuse the static AST, dynamic data flow and function calling relations into GNNs. Ben-Nun et al. (2018) used the Word2Vec model to embed each token. It is encoded with the context of a new conteXtual Flow Graph which coalesces both control flow and data flow generated from the raw LLVM IR. Shi et al. (2020) and Dufour et al. (2007) further adds the dynamic execution state of the program to the program embedding. Though with great recent advancement, GNNs are poor at learning from large graphs due to high cost of the underlying message-passing procedure. Wang et al. (2020) and Hellendoorn et al. (2019) designed a new model GINN and combined GNN with RNN to obtain more comprehensive information with reduced cost. Program representation has been widely used to solve various problems, such as program repair (Wang et al., 2018; Wang & Su, 2020; Tian et al., 2020), bug detection (Dinella et al., 2020), program classification (Lu et al., 2019; Kanade et al., 2020), cache replacement (Li & Gu, 2020; Liu et al., 2020), code similarity detection (Yu et al., 2020a;b). We use program representation to solve the program classification problem and binary similarity detection problem.

## 6    CONCLUSION

To solve multiple categories of program analysis tasks, we constructed program embeddings based on the assembly code of the program. The semantics of the program source code itself as well as the features of the program compilation can be captured simultaneously. Our model takes full advantage of the multi-category graph structure information present in the program and combines it with semantic information. By testing on two tasks, program classification and binary similarity detection, we obtain 82.58% and 83.25%, respectively.

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
