# OpenReview forum: "Exploring General Intelligence of Program Analysis for Multiple Tasks"
_ICLR.cc/2022/Conference — ICLR 2022 Submitted_

### Official Review · Reviewer_Cbj6 · 2021-10-29

**Correctness:** 2
**Technical Novelty And Significance:** 2
**Empirical Novelty And Significance:** Not applicable
**Recommendation:** 3
**Confidence:** 4

**Main Review:**

Strengths:

+Addresses a challenging problem of building a model to achieve better performances in both compilation-specific and source-code level program analysis tasks.

+Designs a method for generating procedure and program level embeddings from multiple graphs.

Concerns:

1. Novelty of the Modeling Approach: The proposed method of generating the procedure-level embeddings of a program including the use of BERT for embedding basic blocks in the CFGs is very similar to [2]. Although the experimental section compares against [2], an attribution to [2] in the methodology would make the contribution of this work clear to the reader. The contribution of the proposed modeling approach seems to be the addition of CG and DFG for program-level analysis, which could be considered as very incremental.

2. Weak Baseline Used for Source-Code Level Tasks: Different from [2], this work proposes a model that performs both source-code and compilation-related tasks. However, a very relevant work [1] that also uses GGNN to model a program graph (based on CFG and DFG) and targets source-code level tasks is not discussed. Any improvements in the proposed approach (by using assembly code and multiple graphs) should be evaluated against [1] or similar work instead of Ins2vec, which I think is not the strongest (or the most relevant) existing approach. Overall, I think a comparative analysis with [1] or similar work is required for establishing the generalizability claim of this work.

Questions/Comments for the Authors:
1. The methodology of pre-training BERT looks very similar to [2], which performs the training only on CFG.  The proposed approach uses three of the four tasks used in [2]. The task not included is “the block inside graph task (BIG)”, which was effective for the model to understand the relationship between block and graph. Is there any specific reason for not using this task?
2. Should we expect a similar gain in performance for procedure-level analysis tasks? Abstracting a program into multiple graphs: CFG, CG, and DFG derived from assembly code may not provide additional information for procedure-level tasks like function name prediction compared to existing methods (e.g., [1]).
3. In Figure 6, it seems the accuracy of OM [2] is lower than the CFG submodel. What are the factors contributing to this change in performance?

Minor Comment:

Allamanis et al. 2016a and Allamanis et al., 2016b are duplicate entries of the same work.

References:

[1] Allamanis, Miltiadis, Marc Brockschmidt, and Mahmoud Khademi. Learning to represent programs with graphs. ICLR, 2018.

[2] Zeping Yu, Rui Cao, Qiyi Tang, Sen Nie, Junzhou Huang, and Shi Wu. Order matters: Semantic- aware neural networks for binary code similarity detection. In Proceedings of the National Conference on Artificial Intelligence and on Innovative Applications of Artificial Intelligence, pp. 1145–1152, 2020


**Summary Of The Paper:**

This paper proposes a program analysis model based on graph neural networks that performs analysis on the assembly code of a program and uses Control Flow Graph (CFG), Call Graph (CG), and Data Flow Graph (DFG) of the program as inputs. The goal is to design a generalized model that can solve both source-code level tasks (e.g., program classification) and compilation-related task (e.g., vulnerability analysis). Program embedding based on the assembly code allows the model to learn compilation-specific features and the use of multiple graphs reflect semantic and structural information.

**Summary Of The Review:**

Contributions are not well-justified. A very relevant work that addresses similar challenges and applies similar approach is not discussed. Demonstration of improvements are shown comparing to a weak baseline instead of more advance and relevant existing works.

---

### Official Review · Reviewer_PoNp · 2021-11-01

**Correctness:** 3
**Technical Novelty And Significance:** 1
**Empirical Novelty And Significance:** 2
**Recommendation:** 3
**Confidence:** 4

**Main Review:**

### Novelty

The idea of fusing feature dimensions in machine learning for code is not new. See https://arxiv.org/ftp/arxiv/papers/2109/2109.03341.pdf on composite program graphs, or https://arxiv.org/abs/1907.02136 on different types of program features.

### Terminology

Authors need to be more precise about the terms they use. I know this is a learning conference, still, it is imprecise to call program classification, duplicate code detection or the other kinds of applications program analysis tasks. Program analysis in PL refers to data-flow analysis, pointer analysis, etc. if we talk about static analysis.

### Others
(1) In page 4, the Eq.1. should use u in f to represent the neighbor of v. But you used v incorrectly.
(2) In Figure 2, the meaning of the nodes in Subfigure of DFG is not clear, and the variables or instructions represented by each node should be specified.
(3) In the experimental setup, the author introduced the POJ-104 data sets, but did not introduce the code size of the program in the data set. As mentioned earlier, the DFG of the source code contains a large number of nodes and edges, but did not compare the number of nodes and edges generated by the DFG in my work.
(4) In the program classification task, the author proves that he can better solve semantics-based problems by comparing with Ins2vec. However, this comparison cannot reflect the advantages of doing this work on assembly code, because both of the program representation and model are different.
(5) On page 8, row 4, how is 8.6% calculated?
(6) In the program classification task, there is no explanation why the effect on gcc is worse than that in llvm.
(7) How to calculate similarity in binary similarity detection task?
(8) Clone detection is usually performed at a function or a more detailed level, and what is the practical significance of detecting the binary similarity of the entire program.
(9) The author has used CFG and DFG respectively, have you considered the effect of using PDG?
(10) The graphical representations(come from angr), bert, and ggnn used in this work are all existing works, the author just applied them to the assembly code. What is the difficulty of learning semantic information from assembly code compared to source code?






**Summary Of The Paper:**

This work proposes a new approach to solving semantics-based problems and compilation-related problems in program analysis by leveraging graph neural networks from the level of assembly code.

**Summary Of The Review:**

Overall, I think the paper does not pass the novelty test.

---

### Official Review · Reviewer_r5sH · 2021-11-04

**Correctness:** 1
**Technical Novelty And Significance:** 1
**Empirical Novelty And Significance:** 1
**Recommendation:** 1
**Confidence:** 4

**Main Review:**

Sadly, I'm not convinced this approach is the right one for "general intelligence" on program analysis. In fact, I would argue, it's likely almost exactly the opposite of the right approach for reasoning about programs.

If we consider the recent work in the space of program reasoning, we see a definitive body of research moving away from lower levels of intermediate representations (IRs) of code (such as assembly or complier IRs, like LLVM) to richer, higher-order representations like the Intel Labs/MIT/Georgia Tech's context-aware semantics structure (CASS), Berkeley and Facebook's simplified parse tree (SPT), the contextual flow graph from Neural Code Comprehension, and even formal mechanisms such as those used by Prof. Alvin Cheung in his verified lifting work that lifts semantics from code to a higher-order lambda calculus.

The purpose of many of these higher-order representations is it can open the doorway for full algorithmic re-writes, which could result in optimizations that are outside of the scope of traditional compilers (e.g., changing the computational class of an algorithm from O(n^2) to O(n logn), etc.). These types of algorithmic re-writes have not been possible (at least from the 100s of papers I've read in the space of machine programming) using lowering techniques like the ones the authors propose.

While I applaud the authors for investigating this approach; sadly, it has been already fairly thoroughly explored (consider Mike Carbin's work on Ithemal for a concrete example of ISA analysis and the limitations they found in it -- in short, their reach was never beyond basic blocks).

I hope this review is helpful for the authors and I'm sorry that I can't give this paper a vote of confidence. I really like the area of research the authors are exploring, but their approach is not something I am convinced has novel contributions nor am I convinced that the experimental results are compelling. My apologies.

**Summary Of The Paper:**

I believe the purpose of this paper is to try to understanding programs by analyzing assembly code using graph neural networks.

The authors test their system against two program-level tasks and claim an accuracy of 82.58% and 83.25%.


**Summary Of The Review:**

TL;DR: Great area of research, but I found little to no novelty in the paper. Moreover, as a scientist, I believe this approach is likely the opposite of what we should actually be doing for program reasoning. That is we should be lifting program semantics *upwards* to understand their meaning, not lowering programs *downwards* to understand their meaning.

Reject.

---

### Official Review · Reviewer_MnJW · 2021-11-05

**Correctness:** 2
**Technical Novelty And Significance:** 2
**Empirical Novelty And Significance:** 3
**Recommendation:** 3
**Confidence:** 5

**Main Review:**

Strengths
-------------
The paper presents an interesting idea. The usage of multiple representations together is both intuitive and promising.


Weaknesses
-----------------
- The evaluation of the paper is very weak (see my detailed comments)
- No comparison against existing ML-based binary similarity detection tools.
- No comparison across different  hardware architectures
- Scalabailtiy of the technique for large real-world programs is not clear


Overall, even though the idea is interesting, the evaluation section is very weak in the current version of the paper. The authors use an old dataset from 2016 of mostly small programs with only two compilers. Usually, binary analysis tasks like similarity detection get harder with multiple compilers and optimization levels, and across multiple hardware architectures. It is fairly standard in the ML-guided binary analysis papers to evaluate against real-world programs with all such variants. Please see: "TREX: Learning Execution Semantics from Micro-Traces for Binary Similarity" by Pei et al. for a detailed list of related works as well as a much larger real-world dataset. I strongly encourage the authors to use such real-world large datasets, experiment across many compiler/architecture/optimization variants, and compare against different existing baselines as done by Pei et al. I also do not quite understand why figure 3 does not show binary code at all given that in many settings the corresponding source code might not be available? How is the proposed technique supposed to work in such cases?

**Summary Of The Paper:**

This paper presents a graph neural network-based approach to solve two binary analysis tasks (program classification and binary similarity
detection). The key idea of the paper is to merge different forms of representation of binary code (compiler IR, assembly code, etc.).

**Summary Of The Review:**

The ideas are interesting but the evaluation is subpar.

---

### Decision · Program_Chairs · 2022-01-20

**Decision:**

Reject

**Comment:**

The paper presents an approach to neural analysis of programs whose main feature is that it operates on assembly code, therefore can account for issues that depend on things like compiler settings. The other important claim of the paper is that by combining information from the control flow and data-flow graphs extracted from the assembly code, they are able to produce an embedding that can support a variety of tasks.

The meta-reviewer agrees with the reviewers that this paper is not suitable for acceptance at ICLR. The novelty in the approach is quite limited, and the gap between the extremely bold claims of the introduction of the paper and what is actually proven in the experiments is quite significant. The evaluation is not strong enough to be considered state of the art.